# Artificial light at night is a top predictor of bird migration stopover density

Kyle G. Horton [1], Jeffrey J. Buler[2], Sharolyn J. Anderson[3], Carolyn S. Burt [1], Amy C. Collins[1,4], Adriaan M. Dokter [5], Fengyi Guo[6], Daniel Sheldon[7], Monika Anna Tomaszewska[8] & Geoffrey M. Henebry [8,9]

As billions of nocturnal avian migrants traverse North America, twice a year they must contend with landscape changes driven by natural and anthropogenic forces, including the rapid growth of the artificial glow of the night sky. While airspaces facilitate migrant passage, terrestrial landscapes serve as essential areas to restore energy reserves and often act as refugia—making it critical to holistically identify stopover locations and understand drivers of use. Here, we leverage over 10 million remote sensing observations to develop seasonal contiguous United States layers of bird migrant stopover density. In over 70% of our models, we identify skyglow as a highly influential and consistently positive predictor of bird migration stopover density across the United States. This finding points to the potential of an expanding threat to avian migrants: peri-urban illuminated areas may act as ecological traps at macroscales that increase the mortality of birds during migration.

Avian migration represents an intrinsic linkage between diverse systems. While active migration occurs in aerial habitats[1], terrestrial and aquatic stopover locations provide critical sites for migrants to rest[2], refuel, and offer a reprieve from adverse weather conditions[3]. Use of these habitats, whether on land or in the air is anything but random, with some areas showing greater and more consistent use season-after-season, decade-after-decade. An objective and comprehensive understanding of the drivers of migrant activity at the interface of terrestrial and aerial habitats can have wide-ranging ecological and conservation applications—yet such large-scale datasets are currently absent. To this end, we leverage remote sensing data and geospatial tools to quantify avian migrant stopover density across the contiguous United States for spring and fall.

The North American avian migration system is composed of nearly 500 migratory species. Migrants are primarily songbirds from both density and species richness perspectives[4,5]. Among songbirds,

diversity in behavior abounds, including migration phenologies, foraging preferences, and tolerances to anthropogenic change. With species ranging from waterfowl to shorebirds to songbirds, among others, comprehensive and large-scale sampling can be remarkably challenging, especially for songbirds. We currently have a poor understanding of songbird stopover use because of their broad-fronted migration strategy that relies on a distributed patchwork of stopover habitat that covers the entire continental land mass. While community science datasets can capture migrant richness[6] and broadscale phenologies[4], they also capture latent properties of spatial sampling biases[7,8]. Some of these biases can be accounted for statistically[8]; however, it still remains a challenge to quantify active migration to understand migrant turnover. Additionally, with the vast majority of migrants taking flight at night[9], active migration can be challenging to observe visually. Here, the use of weather surveillance radar remote sensing data can inform our understanding of migrant

[1]Department of Fish, Wildlife, and Conservation Biology, Colorado State University, Fort Collins, Colorado, USA. [2]Department of Entomology and Wildlife Ecology, University of Delaware, Newark, Delaware, USA. [3]Natural Sounds and Night Skies Division, National Park Service, 1201 Oakridge Dr., Suite 100, Fort Collins, CO 80525, USA. [4]Conservation Science Partners, Truckee, CA, USA. [5]Cornell Lab of Ornithology, Cornell University, Ithaca, New York, USA. [6]Department of Ecology and Evolutionary Biology, Princeton University, Princeton, New Jersey, USA. [7]Manning College of Information and Computer Sciences, University of Massachusetts Amherst, Amherst, Massachusetts, USA. [8]Center for Global Change and Earth Observations, Michigan State University, East Lansing, Michigan, USA. [9]Department of Geography, Environment, and Spatial Sciences, Michigan State University, East Lansing, Michigan, USA.
✉ e-mail: kyle.horton@colostate.edu

distributions by capturing signals during active migration (i.e., birds in flight).

For more than 75 years, radars have been used to detect birds[10]. More recently, the low-elevation scans of radars have been instrumental in revealing the spatial distributions of migrants as they ascend from stopover habitats at the initiation of flights bound for breeding or nonbreeding grounds[11–13]. The heterogeneity of migrant density aloft within these scans uncovers variability in how migrants are distributed among stopover locations at the ground. Coupled with complementary data such as land cover, vegetation indices, and weather, we can begin to address why migrants stopover in some areas and not others, and start to understand the patterns we see from the immense diversity of avian migrants found in North America.

Such radar applications provide insights into how migrants interact with changing landscapes, like the rapid global brightening of the night sky from anthropogenic artificial light during the Anthropocene[14–16]. Migrants in parts of the United States, Mexico, and Israel have a positive association with artificial lights at night[2,13,17–19]—an attraction that can draw migrants into suboptimal stopover habitats near urban areas, and result in excess mortality. Such human-induced rapid environmental change can bring about ecological traps: a maladaptive habitat choice based on cues that were once reliable to promote fitness[20,21]. While it remains unclear why nocturnally migrating birds are attracted to artificial light, it is clear that migrants are shifting their behaviors in response to this broadscale pollutant[13,22–24]. At times, this pollutant can result in fatal collisions with aerial structures[17]. Yet urban areas can still serve as critical stopover locations, understanding where and when these areas are being used by migrants can help direct mitigating actions[25].

Despite the importance of holistically understanding such systems at large spatial and temporal scales, particularly given rapidly changing climates and landforms, analyses are few. The dearth of studies is primarily due to limitations in access to appropriate data and time series to address these questions. Previous analyses focus on local to regional scales[2,12,13,19] and may be overlooking macroscale patterns, particularly those important for conservation action. Recent large-scale investigations reveal novel anthropogenic barriers to migrant stopover, such as the vast agricultural region of commodity crops grown in the Midwest United States[26].

Using the US NEXRAD network, we harnessed the hierarchical spatial structure of stopover measures from more than 1 million locations, assembled 49 predictors, and amassed 2500 models across the contiguous United States to provide the first view of continent-wide migration stopover. We generated spring and fall bird stopover density layers—predicting that greater forest cover[26], higher levels of artificial light pollution[13,22,27], and higher values of vegetative productivity[28] would result in higher levels of stopover density. Additionally, we developed hotspot maps of relative stopover intensity at the scale of a night's flight distance, with the understanding that conservation decisions, like the mechanics of bird migration, are hierarchical in space and time.

## Results and discussion

In all, we processed 3,066,623 weather surveillance radar scans for this analysis, from which we assembled approximately 133,000 scans that aligned with optimal exodus sampling to maximize stopover density while preserving spatial heterogeneity. During model training, we held out $10 \times 10$ km subsets of our training data to assess model performance. During spring, we found a strong positive correspondence between predicted and observed values ($R^2 = 0.85$, $F_{1, 42430} = 244600$, $p < 0.0001$; Fig. 1A), with strength varying only slightly across three broad flyway groupings (Western $R^2 = 0.81$, Central $R^2 = 0.82$ and Eastern $R^2 = 0.73$). Similarly, fall holdout experiments showed a strong positive relationship between predicted and observed values ($R^2 = 0.87$, $F_{1, 42864} = 287300$, $p < 0.0001$; Fig. 1B), again with some variation across flyway groupings (western $R^2 = 0.78$, central $R^2 = 0.84$ and eastern $R^2 = 0.80$).

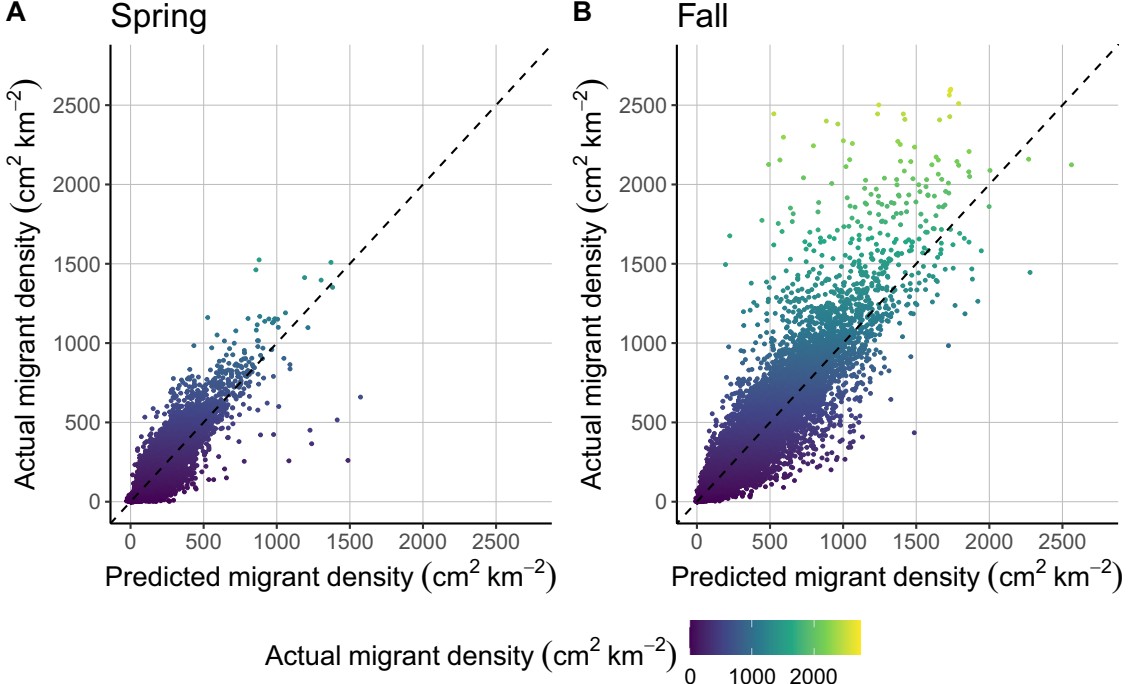

**Fig. 1 | Scatterplot of spring and fall predicted versus actual measured migration stopover density held out during model training from model sets.** Points were held out from a 10-km bounding box within one of the randomly selected 400-km bounding boxes and one year was randomly selected, among the five possible years. The 10-km box was centered on the median X and Y coordinates of training points within the 400-km bounding box. Linear regression fit for **A** spring: $F_{1, 42430} = 244600$, $p < 0.0001$, $R^2 = 0.85$; **B** Fall: $F_{1, 42864} = 287300$, $p < 0.0001$, $R^2 = 0.87$. The dashed line shows a 1:1 relationship between predicted and actual migrant density.

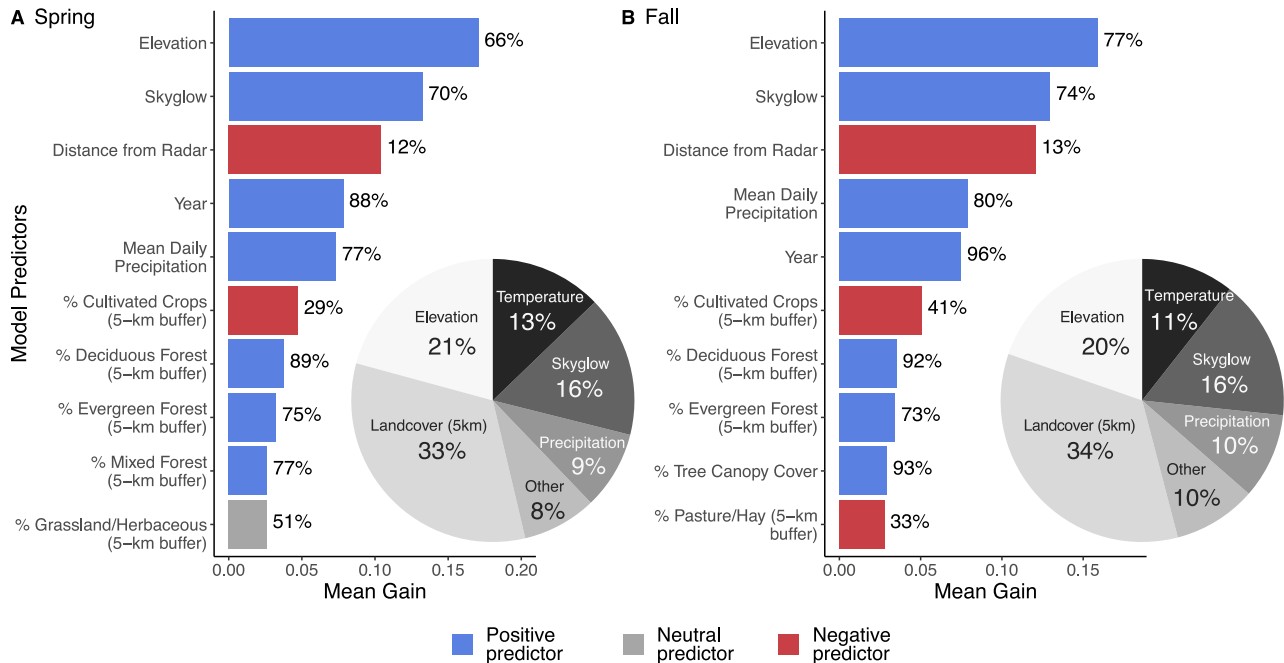

**Fig. 2 | Top-10 variable importance plots ranked by mean gain for spring and fall.** Top-10 variable importance plots ranked by mean gain for **A** spring and **B** fall. Blue bars show predictors that had a positive influence on migrant stopover density in >55% of models, red bars show predictors that had a negative influence on migrant stopover density in >55% of models, and gray shows neutral predictors (between 45% and 55% positive). Note that neutral indicates that the variable was not dominantly positive or negative across all models. However, it could hold a negative or positive value. We show the percent of positive instances within models to the right of each bar. Pie charts show the summed percent of gain across non-sampling predictors categories (i.e., we removed distance to radar and year from the summation). For pie charts, "other" was composed of pooled predictors of landcover at 1-km scale (spring = 1.9%, fall = 2.1%), EVI (spring = 4.0%, 3.4%), percent impervious surface (spring = 0.4%, fall = 0.4%,), and percent tree canopy cover (spring = 2.2%, fall = 3.6%); each individual category accounted for less than 5% of the total gain.

## Drivers of stopover density

From our 2000 400 km scale seasonal models, we found that important predictor variables were similar across seasons, with elevation, skyglow, distance to radar, precipitation, and year making up the top five predictors (Fig. 2). In spring and fall, distance to radar and percent cultivated crop at the 5 km scale showed negative associations with stopover density. The remainder of the predictors in the top ten showed positive correspondence with stopover density. Among the terrestrial predictors, percent tree canopy cover, percent deciduous forest (5 km scale), percent evergreen forest, skyglow, and precipitation most consistently showed a positive association with stopover density, with upward of 70% of models showing positive associations. In spring, of the 47 predictors with sufficient data, 33 showed positive associations, 7 showed neutral associations, and 5 showed negative associations. In the fall, similar patterns were observed, with 32 positive associations, 7 neutral associations, and 8 showing negative associations. Interestingly, in both spring and fall, of the 49 predictors, land cover proportions measured within 1 km pixels ranked the lowest by gain, except for perennial ice at the 5 km scale, which was second to least important in spring and fall.

## Macroscale patterns of stopover density

During spring, we found the central portion of the country showed the greatest stopover densities—in fact, on average, migrant stopover density within this region was 1.5 times greater in the central flyway as compared to the eastern flyway, and 2.9 times greater compared to the western flyway (Fig. 3A). In the spring, Arkansas, Oklahoma, Louisiana, Texas, and Mississippi, showed the greatest mean stopover density, in descending order. The highest stopover densities were in the coastal Gulf of Mexico region, particularly southern Texas. In the fall, stopover density was greatest in the southeastern United States, with Alabama, Tennessee, Arkansas, Mississippi, and Georgia showing the greatest

mean stopover density, in descending order. The greatest fall stopover density resided in the eastern flyway, showing 1.2 times more than the central and 5.8 times more than the western flyway (Fig. 3B).

We also generated relative focal stopover maps parameterized by average songbird flight distances (265 km[29]), with three levels of varying intensity (Fig. 3C, D). Broadly, hotspots often resided near coastlines, geographic barriers (e.g., mountain ranges in Colorado and California), and in regions with large swaths of forest. However, these represent generalizations, and each region showed a specific combination of drivers of stopover density (Fig. 4).

Examining differences in seasonal magnitudes of stopover density (spring vs fall), we found that 70.7% of 1 km pixels showed higher stopover density in fall. On average, pixel-by-pixel, stopover densities were 66% higher in the fall (Fig. 5)—32% of the contiguous US area showed a 100% increase in stopover from spring to fall. Broadly, the greatest positive differences (denoting fall showing higher stopover density) occurred in the eastern half of the United States and the mountain west region. Negative differences (denoting spring showing higher stopover density) were found throughout the coastal western United States, Texas and Louisiana coastlines, and the northern Great Plains region (Fig. 5).

Stopover locations are paramount to the passage of billions of migratory birds. For decades, identifying and prioritizing stopover locations has remained a scientific priority[30]—we fill a perennial gap by providing the quantification of high-density stopover locations. We present the first contiguous US view of stopover density for spring and fall migration seasons, filling broad spatial gaps of previous assessments[26]. Our results provide a comprehensive understanding of macroscale stopover biogeography while delivering high-resolution stopover layers that can be leveraged across scales to fit conservation priorities. We reveal stopover hotspots throughout the contiguous U.S., whereby we identify some of the densest stopover locations

## Absolute migrant density

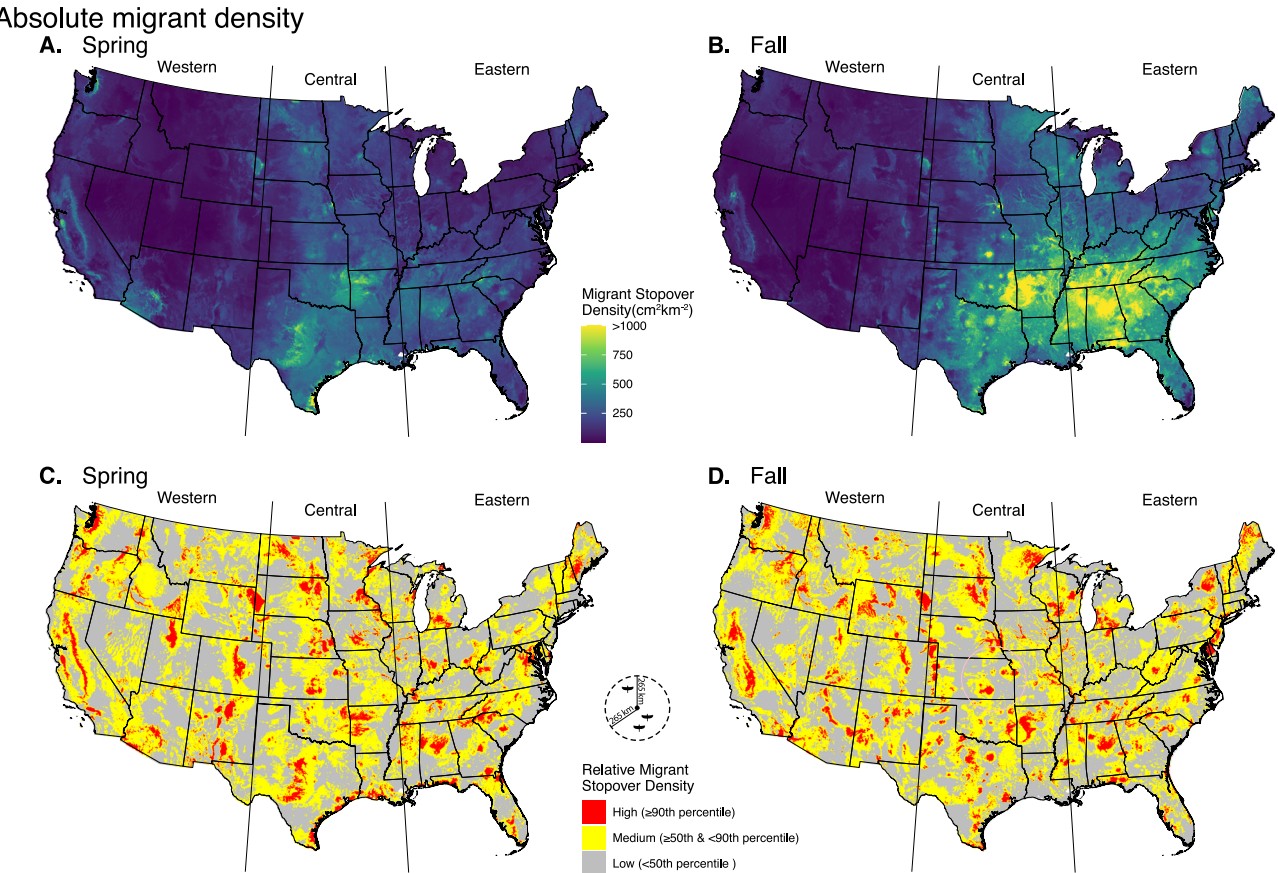

**Fig. 3 | Migratory bird stopover density and hotspot maps for the contiguous United States.** **A** Spring and **B** fall predicted migrant stopover density for 2020. **C** Spring and **D** fall relative stopover categories from predicted 2020 stopover density. Red shades denote pixels above the 90th quantile of predicted stopover density, yellow pixels between the 50th and 90th quantile of migrant stopover density, and gray shows pixels below the 50th quantile of migrant stover density.

Relative quantiles identified using a circular focal window radius of 265 km, which relates to measured average nightly flight distances of tracked free-flying Swainson's (*Catharus ustulatus*) and hermit (*C. guttatus*) thrushes[29]. The western flyway is defined as the contiguous United States west of 103° west longitude, the central flyway is the contiguous United States between 103° and 90° west longitude, and the eastern flyway is the contiguous United States east of 90° west longitude.

residing far from traditional ecological barriers (e.g., coastal barriers). The overall eastward shift in stopover hotspots from spring to fall is consistent with the clockwise looped migration trajectories for migratory birds in North America[31].

### Is avian migration reshaping in the Anthropocene?

We offer maps of relative stopover density analyzed within windows that scale to the distance that migrants generally fly in between stopover locations. Classifying relative stopover hotspots at this biologically based scale reveals a network of important stopover stepping-stones, despite evidence that landbird species generally migrate in a broad front and are ubiquitous during migratory stopover[26]. Our approach also illustrates a different scale for prioritizing stopover areas for conservation, which otherwise would typically occur within a spatial hierarchy that scales with jurisdictional levels (i.e., national, state, local, Figs. 3 and 4, Supplementary Fig. S5). Our maps can help guide conservation efforts to protect critical habitats, and collectively contribute to the full-annual cycle conservation of migratory birds.

Consistently, we found the amount of forest cover, skyglow, and precipitation had positive associations with stopover density. Overall, land cover at the broadest scale (5 km) was the most important class of non-sampling predictors, accounting for 33-34% of summed predictor gain (Fig. 2). Our analyses broadly reveal forest cover measures, whether percent canopy cover or percent forest land cover types (e.g., deciduous, evergreen, mixed forest), are critical drivers of migrant stopover across the contiguous United States. Vegetation canopy

height and canopy complexity are additional potential variables to include in the stopover distribution modeling[32,33]. GEDI (Global Ecosystem Dynamics Investigation), the vegetation lidar onboard the International Space Station, generates products from which to estimate these habitat variables that have been shown to relate to bird diversity and abundance[34–38]. Our stopover modeling framework can be adapted to integrate these additional predictors. Furthermore, more than two decades of US NEXRAD awaits investigation in this capacity, with insights dating back to the mid-1990s possible. Examinations at the interaction of stopover density, stopover timing, and land use change can illuminate how this hemispheric system of migration is being reshaped.

Individually, elevation and skyglow were most important (by predictor gain) and showed the strongest positive slopes across predictor space. Moreover, in the western flyway, skyglow was found to be the top predictor of stopover density. With 28% population loss of migratory birds in the last five decades, and many regions becoming drier[39], less forested[40], and brighter at night[15,16], preserving important and ecologically functional stopover locations is ever more important. Understanding how human-induced environmental change affects how and where migrants use stopover habitats is critical for conservation efforts as artificial lights act as an attractant for many avian species, oftentimes with negative consequences including fatal collisions with built structures[41,42], decreased connectivity[43,44], and changes in phenology[45–48]. These findings lend support to the hypothesis that light pollution can act as an ecological trap for migratory birds,

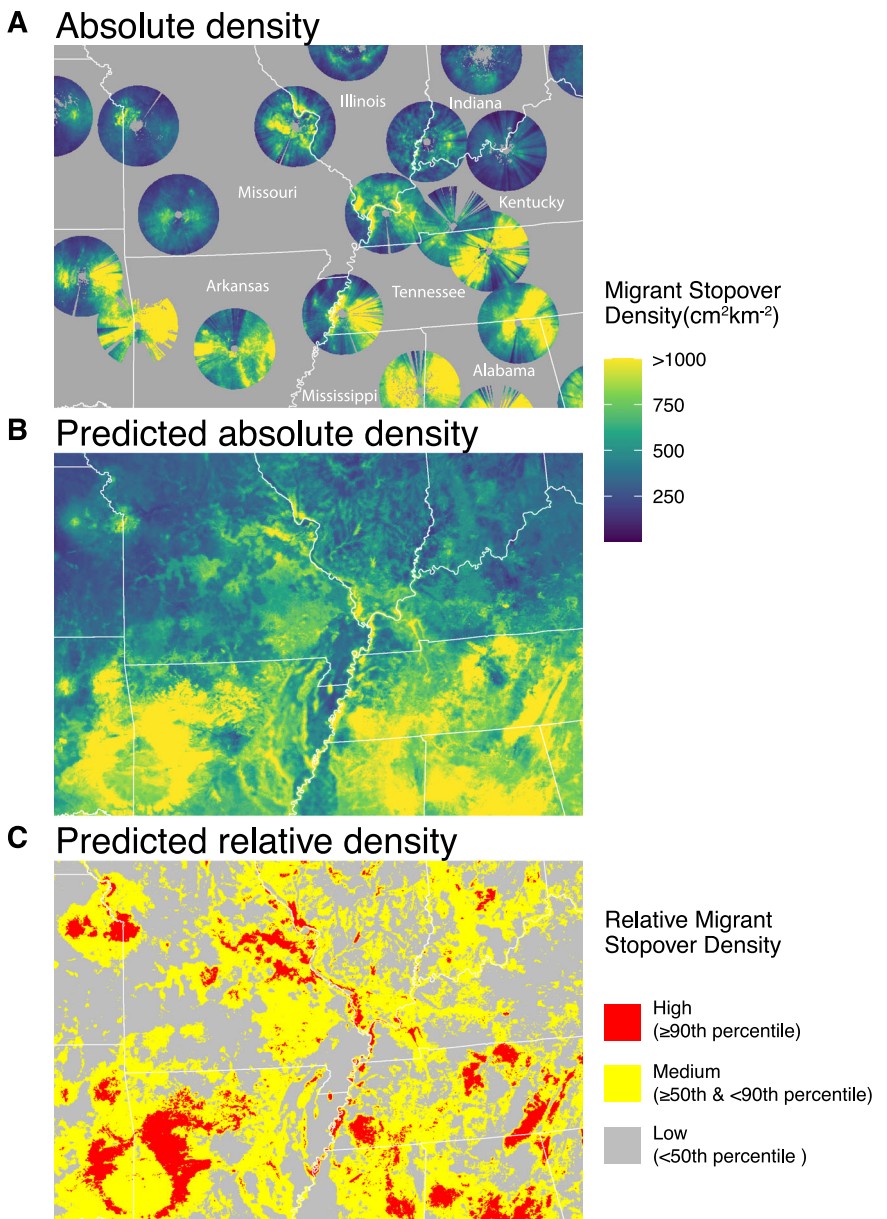

Fig. 4 | Midwest inset showcasing measured and predicted stopover density.
A Measured stopover density (cm²/km²) for Fall of 2020 from 7.5-km to 80-km from radar stations in the Midwest region of the United States. B Predicted stopover density for Fall of 2020. C Relative stopover categories, with red denoting pixels above or equal to the 90th quantile of predicted stopover density, yellow pixels equal to or greater than the 50th and below 90th quantile of migrant stopover density, and gray showing pixels below the 50th quantile of migrant stover density. Relative quantiles identified using a circular focal window radius of 265 km.

drawing migrants into suboptimal stopover habitats and potentially dangerous stopover locations with increased risk for collisions with structures and predation[17,41,49]. However, with skyglow increasing migration stopover density, a challenging conservation dilemma arises. Specifically, are these bright areas, which may fit our definition of a stopover hotspot, a result of attraction from artificial lights or important ecological regions, or the combination of the two? Further insight could be gained by integrating migration passage rates with stopover densities to estimate the percentage of passage migrants that stop in an area (i.e., stopover to passage ratio[2]). A higher stopover to passage ratio for bright areas compared to dark areas could indicate an ecological trap. Currently, bright areas are undoubtedly high-use areas, but if light pollution is mitigated, would stopover density redistribute spatially?

In a world increasingly characterized by changing habitats and climate, it is critical to understand drivers of migrant distributions−

especially at critical stopover locations paramount to their success. With skyglow growing at over 10% per year in North America[15], and its broad and consistent importance in predicting migration stopover− broadscale collaboration, advocacy, and development of lighting policies will be necessary to reverse the rise of this global pollutant[50]. Yet while our results yield a first continental-scale perspective of this ecological threat, our understanding of light pollution and its impacts on avian migrants is far from complete−basic mechanisms of why migrants are attracted to lights remain at large.

## Methods
### Radar data download
The United States National Weather Service, Federal Aviation Admin-istration, and Air Force jointly run the Next Generation Radar (NEXRAD) network. This network is composed of 159 S-band (10-cm wavelength) radars, 143 of which reside in the contiguous US, each collecting 360°

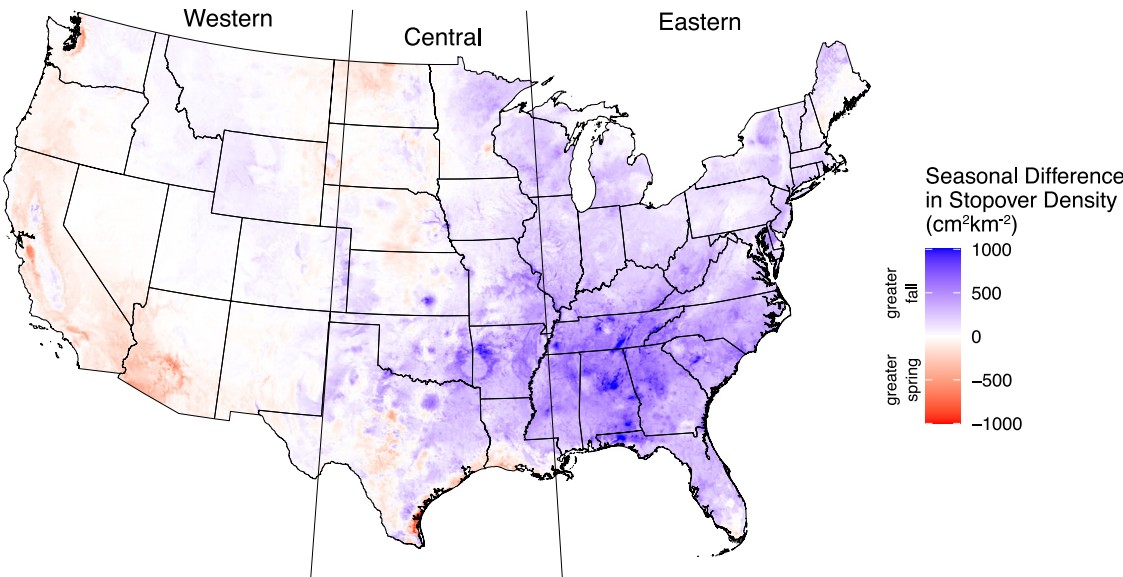

**Fig. 5 | Seasonal difference in predicted migrant stopover density (Fall minus spring stopover density).** Blue shades show greater stopover activity in the fall, red greater activity in the spring, and white showing no seasonal difference.

horizon (azimuthal) scans of the lower atmosphere every 5–10 min, starting at an elevation angle of -0.5°, and increasing in elevation angle with successive scans, reaching up to 19.5° depending on the volume coverage pattern[51]. Since 2008, these radars have operated in a super-resolution format, generating polar grids (plan position indicators, PPI) with 250 m range resolution and 0.5° azimuthal resolution. Individual sampling volumes (hereafter voxels) possess dimensions of 250 m × 0.5° × 1.0°. In this study, we focused on radar reflectivity ($\eta$), a measure of the amount of radiation reflected back to the radar by objects in the air that scales with animal density[52–54]. We quantified migrant stopover density from reflectivity for 2016 to 2020 in spring (93 nights from March 15 to June 15) and fall (93 nights from August 15 to November 15). We downloaded all radar scans from sunset to 2.5 h after local sunset from the Amazon Web Services repository (https://s3.amazonaws.com/noaa-nexrad-level2/index.html). While our analysis focused on one instantaneous measure per night, a priori, the sampling instant that standardized the nightly site-specific timing of stopover exodus was not known, so a wide window of scans was initially downloaded and processed (details below).

**Radar clutter mitigation**
For each downloaded scan, we first worked to remove non-biological clutter and contamination. For every scan, we removed rain at the voxel level using the MistNet algorithm[55], a convolutional neural network that makes binary discrimination between biological or precipitation-contaminated voxels using radar measures of reflectivity, radial velocity, and spectrum width. For every voxel identified as precipitation-contaminated, we set reflectivity values to NA. Because voxels containing precipitation can result in considerably stronger echo returns than biological targets, they pose a greater risk of biasing our results—for this reason, we took a conservative approach and also set voxels within 5 km of precipitation to NA. We removed scans if greater than 30% of the PPI (from 7.5 to 80 km) was precipitation-contaminated.

We removed topographic and other ground-based clutter using two static binary masks (clutter/not clutter). First, we generated radar-specific topographic clutter layers based on underlying elevation and beam geometry for the 0.5° elevational scan. If the radar beam intersected the terrestrial landscape, we labeled it as a contaminated voxel, and it was permanently removed from further analysis (set to NA). Second, we generated radar-specific layers of contaminated voxels

characterized by consistently high reflectivity measures throughout the life span of the radar. For each station and year from 1995 to 2020, we computed the probability of detection (POD) for 30 clear scans from the month of January. We processed the first 300 January scans starting on January 2 00:00 UTC to avoid New Year's Eve disturbances. For each scan, we computed reflectivity on a fixed polar grid of 1° by 500 m to a range of 150 km at elevation angles 0.5°, 1.5°, 2.5°, 3.5°, and 4.5° with values clipped to a ceiling of 35 dBZ. We then selected the 30 scans with the lowest total linear reflectivity as representing clear conditions. From these scans, we computed POD at 10 dBZ, which is the fraction of scans for which the reflectivity exceeded 10 dBZ. A grid cell was marked as clutter if POD at 10 dBZ was 20% or more in two or more years: this step helped mitigate false positives due to actual atmospheric phenomena in a single year. These voxels could contain clutter from terrestrial structures such as—but not limited to—buildings, broadcast towers, trees, and wind turbines.

In some regions of the southern United States, high-density bat emergences from roosts also coincide with the onset of nocturnal bird migration. Because large bat roost departure events show unique ring-shaped features that are visually obvious[56], we were able to screen for these signatures and generate buffers around roost locations and also permanently remove these areas from further analysis (set to NA). In all, we screened 12 sites (KAMA, KCRP, KDFX, KEOX, KEWX, KGRK, KHGX, KLBB, KMXX, KSJT, KTLH, KVAX); states included Alabama, Florida, Georgia, and Texas. One site, KEWX (San Antonio, Texas) was so severely contaminated by bats that we removed the station completely from our analysis, reducing the number of radars to 142. Lastly, we note that insect classification remains a challenging task at the voxel scale within NEXRAD observations. However, because radar reflectivity is proportional to the product of the scatter diameter to the sixth power, measures strongly skew toward larger scatterers, like birds[57]. Additionally, past studies on NEXRAD data, such as ref. [58] showed no significant difference in models of bird migration intensity trained on data with or without insect-dominated scans via airspeed filtering. For this reason, we are confident that our measures reflect signals from bird targets.

**Radar range correction**
Because the narrow radar beam samples at a tilted angle (lowest -0.5°), the beam, and by extension airspace sampled, is higher with increasing

range from the radar and only samples a fraction of the birds above the ground. This characteristic tends to result in fewer birds detected with an increasing range from the radar and raises concern when trying to map biological activity across large spatial extents[59]. To reduce this bias, we used a range-correction technique whereby we first calculate the vertical distribution of migrants by generating a vertical profile of reflectivity (VPR)[60] between 0 and 2000 m above ground level using 100 m altitude bins. For any voxel, the profile allows us to predict the amount of reflectivity through a column of airspace as if it were entirely sampled by the radar. We used the *integrate_to_ppi* function in the R package bioRad to generate PPIs of the vertical integration of biological scatterers based on all available elevation scans[61,62]. We corrected for range effects due to partial beam overlap with the layer of biological echoes at larger distances from the radar. We restricted our range correction to voxels with adjustment factors less than 10 and only to a maximum range of 80 km.

### Radar exodus selection time

Various approaches have been used to select radar scans to align best with the sampling of migrant take-off. If scans are selected too early in the night, they are likely to represent an underestimate of migrant activity; however, if too late in the night, the linkage between the spatial heterogeneity in radar signatures and the underlying terrestrial landscape is eroded (i.e., migrant distributions are largely homogeneous). This tension makes selecting the sampling time particularly important. While some early radar studies selected scans at specific sun angles[12,60,63] (e.g., 5.5°, 7.0°, 8.0°, 9.0°), others have found geographic variation in flight initiation times[13]. Because our study examined a spatial scale larger than any previously undertaken, it was imperative that we worked to capture spatial variation in exodus times, rather than selecting a fixed angle for all sites. To model exodus times, we used radar scans from sunset to 2.5 h after sunset, and for each scan, we calculated the median radar reflectivity (η) of the lowest elevational scan (-0.5°) out to 100 km. We fit a generalized additive model (GAM) to median η (response) with the hour after sunset as a smooth predictor (k = 10) and date as a random effect. We then used this model to determine the time after sunset that predicted the greatest rate of change in η. Across all radar stations, this analysis resulted in an average sampling time of 49 min after local sunset but ranged between 27 and 77 min after local sunset (Supplementary Fig. S1). Using these times, we explored drivers of variation in sampling time using two linear regression models with sampling time as the response variable and elevation and mean skyglow as predictors. We quantified mean skyglow at two scales, within 37.5 km and 80 km buffers; one model was run for each scale. At both scales, we found that radar site elevation was a significant predictor of sampling time ($p < 0.001$), but not mean skyglow (37.5 km buffer, $p = 0.383$; 80 km buffer, $p = 0.751$). These models explained 51.7% (37.5 km scale) and 51.4% (80 km scale) of the variance in sampling time.

Using site-specific exodus sampling times, we selected and assembled all seasonal range-corrected and filtered scans closest to the period of interest. With scans assembled, we took the mean of stopover density across all sampling nights within a season-year interval. For each season-year combination, we mosaicked stopover densities from all 142 radar stations, taking the mean where overlap in sampling areas occurred. Lastly, we resampled to a 1 km resolution. These data served as our response variable for stopover modeling.

### Predictor variables for niche model

To understand the drivers of stopover densities, as estimated by radar, we assembled a broad suite of predictor variables, including the enhanced vegetation index (EVI) (4 predictors), land cover classes and composition (30), percent canopy cover (1), percent impervious surface (1), accumulated nocturnal degree-days (8), precipitation (1), skyglow (1), elevation (1), distance to radar (1), and year (1). In all, we

used 49 predictors, of which 48 were geospatial (see Supplementary Fig. S2). We use National Land Cover Data (NLCD) to capture cover type associations, MODIS data to quantify annual and within-season fluctuations in vegetation greenness and surface temperature, and VIIRS DNB to derive an index of skyglow.

**MODIS enhanced vegetation index.** We used the MODIS/Terra vegetation indices monthly L3 global product (MOD13A3 V061) at 1 km spatial resolution[64]. We downloaded the 14 tiles covering the contiguous US from 2016 through 2020. We focused on two scientific datasets—the monthly EVI and the monthly VI Quality. We filtered pixels based on values in the quality bits. We included those pixels with (1) VI Quality equal to 00 or 01; (2) VI Usefulness from "highest quality" to "decreasing quality" (0000 to 1010); (3) Aerosol Quantity from "low" to "intermediate"; (4) Possible snow/ice equal to No (0). We then merged all tiles and reprojected the data into the Albers Conical Equal Area projection with 1 km pixel resolution to match other predictor variables and estimates of stopover density using nearest neighbor resampling. We segmented the EVI into four spring monthly periods (March to June) and four fall periods (August to November). We included each monthly layer as an individual predictor (four per seasonal model).

**MODIS land surface temperature (LST) products and calculation of thermal time.** We used the MODIS/Terra and MODIS/Aqua land surface temperature/emissivity products (MOD11A2/MYD11A2 V061) at 1 km spatial resolution which provide the 8-day average of Land Surface Temperature (LST) from all qualifying MOD11A1/MYD11A1 LST pixels[65,66]. Using the same 14 MODIS tiles, we selected two scientific datasets from each product: LST_Night_1km (8-day nighttime 1 km grid Land Surface Temperature), and the corresponding QC_Night (Quality control for nighttime LST and emissivity). We filtered pixels using values in the quality bits of each product. We included those pixels with Mandatory QA flags equal to 00 or 01 and LST Error flag (all bits for average LST error <3 K). We then converted LST from K to °C.

We chose to focus on nighttime LST for two reasons: (1) daytime LST exhibits a troublesome positive bias relative to near-surface air temperature while nighttime LST does not[67], and (2) it corresponds more closely to nocturnal bird migration patterns than daytime LST[58]. To characterize the progression of thermal time during the year, we calculated Accumulated Nocturnal Degree Days (ANDD, accumulated only when the average nighttime degree-days were above the base temperature of 0 °C[68]). We used a modification[69] of an algorithm used in earlier studies[67,70]. The transformation of two nighttime observations from Terra and Aqua into minimum MODIS LST used [1]:

$$\text{Minimum MODIS LST} = \min(\text{LST}_{2230}, \text{LST}_{0130}) \qquad (1)$$

Where $\text{LST}_{2230}$ is the nominal nighttime overpass of the Terra MODIS and $\text{LST}_{0130}$ is the nighttime overpass of the MODIS on Aqua[69]. To fill gaps due to missing or excluded pixels in quality filtering, we used Seasonally Decomposed Missing Value Imputation[71]. It first removes the seasonal component from the time series, then performs imputation on the deseasonalized series using the weighted moving average with k = 4 (8 observations with 4 left and 4 right), determines the exponential weighting based on the deseasonalized series, and then adds back the seasonal component.

We further generated Nocturnal Degree Days (NDD) dataset, where we filtered out minimum MODIS LST below 0 °C at compositing period t as the maximum of minimum MODIS LST and $T_{base}$, which was set to 0 °C [2]:

$$\text{NDDt} = \max((\text{minimum MODIS LST}_t - T_{base}), 0) \qquad (2)$$

Since we used the 8-day product, we multiplied NDD composite values by 8 to account for that 8-day compositing period. However,

because we work on a monthly time frame, we did the multiplication based on the actual number of days per month (between 0 to 8 depending on the compositing periods alignment with the calendar). We checked each 8-day composite for whether it spanned across months. For example, composite DOY 057 starts on 02/26 and ends on 03/05, so that composite covers three days in February and five days in March (and shifted in leap years). If the composite was in the middle of the month, then multiplication was by 8. We then accumulated across the year using [3]. Accumulations were reset to zero at the start of each year.

$$ANDD_t = ANDD_{t-1} + (NDD_t \times m) \tag{3}$$

where $m$ is a specified number of days [from 0 to 8] within a month from 8-day composite.

Finally, we merged all tiles and reprojected data into the Albers Conical Equal Area with 1 km pixel resolution using nearest neighbor resampling. We segmented ANDD into eight spring (March to June) and eight fall (August to November) periods based on the first and second half of each month. We included each semimonthly layer as individual predictors.

**National Land Cover Database.** We used the 2016 and 2019 National Land Cover Database (NLCD) releases to characterize the percent canopy cover (2016), percent impervious surface (2016, 2019), and land cover classification (2016, 2019). NLCD products are distributed at a 30 m resolution and land cover products contain 16 unique land cover classes. To simplify the land cover classification scheme, we merged Developed, Open Space (class 21) and Developed, Low Intensity (class 22) into one Developed class. All three datasets (landcover, percent impervious surface, and percent canopy cover) were resampled to 25 m, using a nearest neighbor rule, allowing each pixel to neatly delineate within the target 1 km resolution. Then, within a 1 km pixel, we calculated (1) for both percent canopy cover and impervious surface, the average percent of each, (2) for modified land cover, the percent of each cover type. The frequency of each land cover class was included as an individual predictor variable, yielding a total of 15 variables. Lastly, to characterize the neighboring region surrounding the 1 km pixel of interest more broadly, we also calculated the percent of each cover type within a 5 km buffer. Again, each land cover class represented one predictor variable. In total, we derived 32 predictor variables from the suite of NLCD products. For landcover and percent impervious surface, we assigned radar data from 2016, 2017, and 2018 to the 2016 NLCD products, and 2019 and 2020 to the 2019 NLCD products.

**Precipitation.** We used Daymet[72] version 4 R1 to calculate the mean daily precipitation for spring and fall for all years of interest (2016–2020). Daymet is distributed at a 1 km resolution gridded across North America. We reprojected the 1 km grid to align all pixels to previously resampled spatial layers.

**Skyglow.** We used the Visible and Infrared Imaging Suite (VIIRS) Day Night Band (DNB) monthly cloud-free DNB composite products to calculate skyglow. These data products are produced by the Earth Observation Group, Payne Institute for Public Policy, at a 15 arc second, or roughly 500 m resolution. These products remove non-stable lighting from the imagery[73]. The average monthly composites for spring were weighted averages for the months of March to June of each year and for the fall season August to November for years 2016 to 2020. We used the "vcmsl/vcmslcfg" (VIIRS Cloud Mask-Stray Light Removed) monthly VNL V1 as the base data for the simplified all-sky light pollution model (sALR)[74]. To calculate skyglow, the sALR model was run for each seasonal image from 2016 to 2020. This simplified spatial model provides a high-confidence estimate of an all-sky light pollution ratio (ALR) metric for large regions.

The all-sky light pollution ratio (ALR) is the ratio of artificial light at night to the natural night sky[74]. For example, a ratio of 0.33 means that the sky is less than 33% brighter than the natural night sky which is pristine (ALR = 0.0); whereas a ratio >10.0 indicates that the Milky Way is invisible. We used skyglow, rather than VIIRS DNB to capture indices of light pollution that may be perceived on the horizon by in-flight migratory birds, rather than measures from VIIRS DNB, which capture upward radiance.

**Elevation.** We used the NASADEM data product (NASADEM_HGTv001) to capture elevation throughout the contiguous United States. The NASADEM is derived from telemetry data from the Shuttle Radar Topography Mission (SRTM), a collaboration between NASA and the National Geospatial-Intelligence Agency (NGA), as well as participation from the German and Italian space agencies. We resampled the original 1-arc second spatial resolution to 1 km.

### Correlation of predictors

When examining the influence of model predictors, we wanted to understand the relationship between a suite of core predictors, including skyglow and habitat cover types. While we predict skyglow, forest cover, and riparian corridors to have a positive association with stopover density, we wanted to understand if their occurrence was correlated (e.g., is high skyglow also associated with a high percent of canopy cover?). To test this, we summarized the distribution of correlation coefficients and proportion of significant correlations (based on an alpha value of 0.05), examining pairwise correlations of skyglow and % canopy cover, skyglow and proportion of forest cover types (NLCD classes 41, 42, and 43), skyglow and proportion of open water (NLCD class 11), and just as a proof-of-concept, % canopy cover and proportion of forest cover types (NLCD classes 41, 42, and 43), which we predicted would be highly correlated. For habitat-specific correlations, we focused on the 5 km buffer scale. We used predictors from one season, 2016, because % canopy cover only had one replicate in our study—otherwise, in our correlations, pixel values would be pseudo-replicated (i.e., the same value repeated in 2017–2020). Lastly, rather than examining correlations across all 2016 pixels (~1 million), we examined Pearson's correlation coefficient from 10,000 random draws of 100 locations. We show these results in Supplementary Fig. S4. Generally, we found very weak correlations between skyglow and canopy cover, skyglow and forest cover types, and skyglow and open water (median Pearson's correlation coefficient values between −0.1 and 0.04), with between 1.2% and 10.2% of correlations showing significance. We found strong positive correlations between % canopy cover and proportion of forest cover types, with 100% of correlations showing significance.

### Niche modeling

In total, we included 49 predictor variables in our modeling of migrant stopover density. We extracted 1,002,511 random points from the radar coverage area (50.5% of possible locations) and ensured that no single location was replicated in our training dataset (e.g., checking that multiple random points did not fall within a 1 km pixel). Our radar coverage area represented 26.2% of the contiguous United States; distributional modeling filled the remaining 73.8% of the land area. We used gradient-boosted trees, carried out through the XGBoost[75,76] package in R, to examine relationships between predictor variables and stopover density. This approach uses a tree ensemble model, which consists of a set of regression trees, as applied in a supervised learning environment that relates a training dataset to a response variable (i.e., stopover density). We divided our dataset into three groups: a training set (75%) for learning; a validation set for model tuning (15%); and a test set to evaluate performance (10%). We randomly assigned locations to these categories to ensure independence across the sets at the location level.

We guarded against overfitting the model by choosing the best combination of parameters, iterating through variations of the following parameters: *max_depth* = 8, 10, 12, 14, 16, 18, 20, *min_child_weight* = 1, *gamma* = 0, 1, 2, 5, and 10, *colsample_bytree* = 1, and *subsample* = 0.7 and 1.0. We set the learning rate to 0.1 and set *early_stopping_rounds* to 10 to determine the optimal number of boosting iterations for that learning rate. We examined $R^2$ on the test set and settled on a *max_depth* = 16, *min_child_weight* = 1, *gamma* = 0, *colsample_bytree* = 1, and *subsample* = 1.

To prevent long-distance learning in these models[7], we spatially partitioned our data by generating 2,500 random locations across the contiguous United States. Long-distance learning occurs when associations between response and predictors are learned in one region and are then used to make predictions in a distant spatial environment. This approach has been used for modeling species distributions from eBird community science data[7], but never for weather surveillance radar data. For 2000 of these locations, we subset our training data to within a 400 km bounding box centered on the random point (Supplementary Fig. S3). For the remaining 500 random locations, we subset our training data to an 800 km bounding box centered on the random points to provide broader coverage, particularly in areas with fewer radars (Supplementary Fig. S3). For each model generated, we used gradient-boosted trees to predict stopover density at 1 km pixel resolution within the bounding box. For these predictions, we fixed distance to radar at 35 km and used 2020 predictor variables where possible (e.g., NLCD land cover from 2019). We mosaicked all predictions together to generate a continuous seasonal surface of stopover density, averaging predictions where redundant predictions occurred. From predicted surfaces of stopover density, we also calculated a focal window analysis of relative intensity. We used a circular focal window radius of 265 km, which relates to measured average nightly flight distances of tracked free-flying Swainson's (*Catharus ustulatus*) and hermit (*C. guttatus*) thrushes[29]. For each pixel at the center of a focal window, we identified three levels of intensity: high (greater than or equal to the 90th percentile), medium (greater than or equal to the 50th and below the 90th percentile), and low (below the 50th percentile). Because conservation decisions of land management tend to be hierarchical, we also generated state-level maps of hotspots following the same definitions of low, medium, and high-intensity regions. Lastly, for summaries, we also examined migrant stopover density by flyway classification[77], in which the western flyway is defined as the contiguous United States west of 103° west longitude, the central flyway is the contiguous United States between 103° and 90° west longitude, and the eastern flyway is the contiguous United States east of 90° west longitude.

### Variable importance and directionality

For each model trained, we output variable importance by monitoring gain, which represents the fractional contribution of each feature to the model based on the total gain of a feature's splits[76]. Additionally, for all variables and for each model, we generated partial dependence plots. Partial dependence plots are predictions of the response variable while holding all variables at their median levels, but allowing the variable of interest to vary across the range of values in the training dataset. For both the spring and fall 400 km model sets, we calculated the mean gain for each variable. Additionally, to capture the directionality of how the variable of interest influenced stopover density, we fit a linear model to the predictions from partial dependence plots (*sensu*[8]). For each of these linear models, we extracted the slope coefficient and the *p*-value. We then subset those variables that had *p*-values less than 0.05. Of the linear fits with significant coefficients, we calculated the percent of positive coefficients. From these data, we considered a variable to have a global positive influence if greater than 55% of models showed a significant positive association; we considered the variable to show a negative association with stopover density if less than 45 percent of significant coefficients showed a positive association.

### Holdout experiment

To understand how our models would perform in areas not sampled by the radar, we conducted a series of holdout experiments. We separately reran all 2000 400 km bounding box models, but this time holding out a small spatial portion of data. For each of these models, prior to training, we determined the median of the X and Y positions of the training data and subset a 10-km bounding box centered on that median location. These data were held out of the model training process, and when training was completed, predictions were made at these locations. This holdout approach ensured that training and validation data were spatially separated, which could otherwise inflate our performance metrics due to spatial autocorrelation. We only included predictions in our assessment if at least 25 sampling points resided in our 10 km bounding box, which restricted this experiment to 1324 models in spring and 1340 in fall. We sped up training by increasing the learning rate from 0.1 to 0.25 and injected additional randomness into this model by setting the *colsample_bytree* parameter to 0.75 and the *subsample* parameter to 0.5. We averaged any predictions that were generated for the same location and same year; this could occur if the 10 × 10 km holdout location overlapped in multiple models (e.g., location X from models 1 and 2). Lastly, for each season, we randomly selected one prediction from the five possible years—this step ensured we did not include multiple predictions from the same location (i.e., to limit pseudoreplication). We then compared estimated stopover density versus predicted stopover density.

### Reporting summary

Further information on research design is available in the Nature Portfolio Reporting Summary linked to this article.

## Data availability

The weather surveillance radar and predictor data used in this study are available in the FigShare database under https://doi.org/10.6084/m9.figshare.24438280.

## Code availability

The code used to model our results in this study is available in the FigShare database under https://doi.org/10.6084/m9.figshare.24438280.

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

## Acknowledgements
Funding for this project was provided by NASA Biodiversity 80NSSC21K1143 to K.G.H., M.A.T. and G.M.H. K.G.H. and C.S.B. were supported by National Science Foundation Growing Convergence Research program (GCR-2123405). This material is based upon work supported by the National Science Foundation under Grant Nos. 1661259 (D.S.), 1749854 (D.S.), 1927743 (A.M.D.) and 2017817 (A.M.D.). USDA NIFA Hatch (DEL-00774) to J.J.B.

## Author contributions
K.G.H., J.J.B., M.A.T. and G.M.H. conceived the idea for this paper. K.G.H., J.J.B., F.G., C.S.B., M.A.T. and G.M.H. worked to draft and edit the manuscript. M.A.T. and G.M.H. led the processing and curation of satellite-based remote sensing and geospatial data. S.J.A. led the processing of skyglow indices. A.C.C. assembled precipitation, radar, elevation, and VIIRS DNB data, prepared spatial layers, and screened radar data for bat contamination. A.M.D. and D.S. generated clutter layers. K.G.H. led the processing of radar data, data integration, statistical analyses, and generated figures.

## Competing interests
The authors declare no competing interests.
