## [Peer Review File · Nature Communications]

Artificial light at night is a top predictor of bird migration stopover densityReviewers' Comments:

Reviewer #1:

Remarks to the Author:

The paper by Horton is an impressive continent-wide study of spatial and temporal variation in the density of nocturnal migratory songbirds at daytime stop-over locations throughout the contiguous USA. It uses weather radar data to measure the intensity of birds in the airspace immediately above these sites at dusk (as they take-off from their daytime roosting/refueling sites), as a proxy for the number of birds spending the day in different areas, which seems an appropriate way to measure this at very large spatial scales. The study compares spring and autumn migration across the US, and then models the density of birds at stop-over locations with respect to a variety of predictor variables covering a range of environmental features. Due to the serious declines in migratory bird populations, the fact that migration has high mortality risk, and the importance of stop-over habitat for successfully completing migrations, this is an important and timely topic for study.

The paper finds that stop-over density varies both spatially and temporally, and the outputs can be used to identify important regions for conservation focus. Landcover type, elevation and skyglow are the predictors that best explain this variation. The results appear robust and of interest, though I have some concerns about how the paper is framed, and whether the main message really represents the results. I expand on this point below, as well as mentioning some other minor issues. I believe the paper will be publishable after these issues have been satisfactorily dealt with.

Major comment:

The title of the paper, and (to a slightly lesser extent) the abstract, indicate that this study is all about the impact of skyglow – after reading these, the reader is led to believe that: (i) light attracts the migrants to these stop-over locations; (ii) these locations are otherwise sub-optimal, and act as ecological traps; and (iii) consequently, this results in lower survival and/or fitness of the migrants. However, as lines 186-205 of the results/discussion clearly discuss in a more nuanced way, this is conjecture; the positive association between skyglow and density may be a result of above, but may also be because good quality stopover habitat also, coincidentally, has high skyglow. This paragraph discusses how these issues might be unraveled, but I had expected this paper to tackle that issue based on its title and abstract. I therefore feel that, as currently framed, the manuscript needs to be expanded to include these analyses so that an attempt can be made to answer these questions. (Alternatively, the ms can be reframed to accurately reflect the current content, but then more works need to be done to show how this paper differs substantially from the other studies that have already looked at stop-over habitat usage).

Minor Comments:

1. The methods describe how precipitation, clutter and bats are removed from the signal, but what about insects – how are they removed?
2. Fig S1 shows how exodus time varies geographically, is there any pattern here? Is it affected by the level of light pollution?
3. Figure 1 – the dashed line is presumably the 1:1 relationship, the legend should say this; what about adding the regression lines, which probably will have a slope different from 1:1; why are the predicted values consistently lower than observed when density is high?
4. Line 111. The flyways (west, central, east) are not defined, they should be illustrated (see below)
5. Lines 129-138. The text here needs to be related to a figure, which I suspect is Figs 4A & B. Please cite. Also Fig 4 should illustrate the flyway limits, and I think that box plots could be added to show the density values per flyway that is discussed in this section.
6. As Fig 3 not mentioned yet, the order of the figures should be changed. In fact, Fig 3 is not really discussed anywhere – what is it showing? Needs to be documented in the results.
7. Line 152 – I don't think "tantamount" (meaning equivalent to) is the right word to use here.

Jason Chapman

Reviewer #2:

Remarks to the Author:

The authors of the manuscript titled "Reshaping avian migration in the Anthropocene — continental scale attraction to artificial light at night revealed" provided a comprehensive analysis of bird stopovers during spring and fall migration based on five years of data, over the contiguous United States. The analysis generated maps of migratory bird density (spring and summer) and identified positive and negative relationships of bird density to landscape characteristics. The skyglow variable, quantifying light pollution, had the second strongest positive effect on bird density, after elevation. This result is concerning for bird conservation, as collision with urban structures is a known cause of bird mortality during migration. The manuscript is clear and concise and the figures are representative of the main findings of the manuscript.

The data processing and analyses are robust and well-documented, thus the results and conclusions presented in this manuscript are reliable. While I don't have experience with radar data, the analytical steps the authors provided enough details to understand the radar data processing. The modeling component (gradient boosted trees) was also well-documented and designed. I don't have any suggestions for improving the models. I think this study provides novel and valuable insights into spatial patterns of migratory bird density at stopovers that can inform bird conservation and land management.

I only have a few minor editing suggestions.

Line 38: revise "may act as ecological traps at macroscales, bringing migrants into dangerous, modified habitats." as "may act as ecological traps at macroscales that increase mortality of birds during migration"

Line 96: reword as "amassed 2,000 models at 400 km resolution across the contiguous US"

Line 181: references 34 and 35 don't seem to be studies about the relations between vegetation structure and bird diversity and abundance. A Google Scholar search "bird diversity vegetation structure" returned papers that may provide a stronger support for this statement, for example: Bakx, T.R., Koma, Z., Seijmonsbergen, A.C. and Kissling, W.D., 2019. Use and categorization of light detection and ranging vegetation metrics in avian diversity and species distribution research. *Diversity and Distributions*, 25(7), pp.1045-1059.

Carrasco, L., Giam, X., Papeş, M. and Sheldon, K.S., 2019. Metrics of lidar-derived 3D vegetation structure reveal contrasting effects of horizontal and vertical forest heterogeneity on bird species richness. *Remote Sensing*, 11(7), p.743.

Farwell, L.S., Gudex-Cross, D., Anise, I.E., Bosch, M.J., Olah, A.M., Radeloff, V.C., Razenkova, E., Rogova, N., Silveira, E.M., Smith, M.M. and Pidgeon, A.M., 2021. Satellite image texture captures vegetation heterogeneity and explains patterns of bird richness. *Remote sensing of Environment*, 253, p.112175.

Line 406: provide the spatial resolution of this product

Lines 757-758: revise here to remove the double negative. Possible revised text: "Note that neutral indicates that the variable was not dominantly positive or negative across all models."

Reviewer #3:

Remarks to the Author:

The authors present a novel analysis of factors affecting avian migrations in the United States using data collected from a network of 142 Radars. They include a large number of potential explanatory terms in a niche model including the enhanced vegetation index (EVI), land cover classes and composition, percent canopy cover, percent impervious surface, accumulated nocturnal degree-days, precipitation, skyglow, elevation, distance to radar, and year. The authors identified skyglow as a highly influential and consistently positive predictor of bird migration stopover density in 70% of their models.

The paper is very interesting, well written and brings together multiple large remote sensing data sources from across different platforms. I feel the paper would make a nice contribution to Nature Communications and should be highly cited.

I do however have some questions I would like to see clarified before the manuscript can be considered for publication.

The response data (bird stopover density) is derived from a network of 142 radar stations, each sampling a very small region (250-m x 0.5° x 1.0°). My immediate concern is then that the model outcomes and the authors interpretations of them are highly dependent on the locations of these 142 radar stations in relation to the spatial distribution of the predictors. 49 spatial predictors for a response that is replicated in 142 unique spatial locations is very small replication indeed.

I have not seen any description of the predictor variable conditions at each radar station, or any evidenced discussion of how reflective this is of the distribution of these predictors across the wider US. If radar station are predominantly suburban for example, then their may be considerable bias in the dataset, particularly for those located in arid environments. The end result might be that birds tend to stop over where there is more water or forage, and this happens to be locations where radar stations tend to be located, and also where there are military installations and hence more skyglow.

I'm not saying that this is the case, rather that the authors have not evidenced that it is not the case. A simple presentation of the response variable conditions at each radar station compared to the range of those responses across the US should help (hopefully) rule out any source of bias in the analysis.

REVIEWER COMMENTS

Reviewer #1 (Remarks to the Author):

The paper by Horton is an impressive continent-wide study of spatial and temporal variation in the density of nocturnal migratory songbirds at daytime stop-over locations throughout the contiguous USA. It uses weather radar data to measure the intensity of birds in the airspace immediately above these sites at dusk (as they take-off from their daytime roosting/refueling sites), as a proxy for the number of birds spending the day in different areas, which seems an appropriate way to measure this at very large spatial scales. The study compares spring and autumn migration across the US, and then models the density of birds at stop-over locations with respect to a variety of predictor variables covering a range of environmental features. Due to the serious declines in migratory bird populations, the fact that migration has high mortality risk, and the importance of stop-over habitat for successfully completing migrations, this is an important and timely topic for study.

The paper finds that stop-over density varies both spatially and temporally, and the outputs can be used to identify important regions for conservation focus. Landcover type, elevation and skyglow are the predictors that best explain this variation. The results appear robust and of interest, though I have some concerns about how the paper is framed, and whether the main message really represents the results. I expand on this point below, as well as mentioning some other minor issues. I believe the paper will be publishable after these issues have been satisfactorily dealt with.

Major comment:

The title of the paper, and (to a slightly lesser extent) the abstract, indicate that this study is all about the impact of skyglow – after reading these, the reader is led to believe that: (i) light attracts the migrants to these stop-over locations; (ii) these locations are otherwise sub-optimal, and act as ecological traps; and (iii) consequently, this results in lower survival and/or fitness of the migrants. However, as lines 186-205 of the results/discussion clearly discuss in a more nuanced way, this is conjecture; the positive association between skyglow and density may be a result of above, but may also be because good quality stopover habitat also, coincidentally, has high skyglow. This paragraph discusses how these issues might be unraveled, but I had expected this paper to tackle that issue based on its title and abstract. I therefore feel that, as currently framed, the manuscript needs to be expanded to include these analyses so that an attempt can be made to answer these questions. (Alternatively, the ms can be reframed to accurately reflect the current content, but then more works need to be done to show how this paper differs substantially from the other studies that have already looked at stop-over habitat usage).

Thank you for the critical evaluation of the framing of the manuscript. We agree with the reviewer, and we have made the following adjustments to reflect this concern. We cannot envision analyses at this scale that would definitively rule out the alternatives that the reviewer has suggested could be driving these patterns.

Thus, we have altered the title to read as follows:

Is avian migration reshaping in the Anthropocene? — artificial light at night revealed as a top predictor of stopover density

We have also altered the abstract slightly to offer a more nuanced tone to reflect that we have not proven light is the proximate driving force of stopover ecology of migratory birds.

This finding points to the potential of an expanding threat to avian migrants: peri-urban illuminated areas may act as ecological traps at macroscales that increase mortality of birds during migration.

Lastly, and following Reviewer #3 final comment, we explored the correlation between a suite of predictor variables (e.g., skyglow, % canopy cover, % open water, see Figure S4). From these analyses, it does not appear that skyglow is correlated with characteristics of favorable habitat, and if anything, very slightly the correlations tend to be negative between forest cover and skyglow.

Minor Comments:

1. The methods describe how precipitation, clutter and bats are removed from the signal, but what about insects – how are they removed?

Taxonomic classification is always a challenging topic when using weather surveillance radar measures. As noted by the reviewer, we worked exceptionally hard to mitigate contamination from non-bird related sources (e.g., clutter, beam blockage, bats, rain). Insects, however, are a challenging class to remove without creating other biases. In past studies, e.g., see (Van Doren and Horton 2018)), we examined this problem when measuring nightly migration. In that paper we explored how airspeed filtering, a common tool for insect and bird discrimination, would impact our results. Broadly, airspeed filtering is an approach whereby radar samples with airspeeds less than 5 m/s are considered insect-dominated and samples with speeds greater than 5m/s bird-dominated. This step is a non-trivial task, but in the end, we observed no change in our findings. (Van Doren and Horton 2018)) removed altitude bins with mean airspeed $\leq 5 \text{ m s}^{-1}$ to limit the inclusion of flying insects in our dataset. Predictions made by their model trained without airspeed filtering corresponded closely to those made by the final model with airspeed filtering (Pearson's $r = 0.995$). Prediction error for these two models was comparable (RMSE = 1.422 with filtering; RMSE = 1.442 without filtering).

In our study, we opted not to attempt insect filtering for four reasons. First, past studies on NEXRAD data, such as Van Doren and Horton (2018) showed no significant change. Second, insect filtering techniques for stopover ecology are not suitable to remove partial scans (e.g., MISTNET precipitation algorithm can filter individual pixels), but rather whole scans, and thus for stopover modeling, results in the removal of whole nights. This type of filtering can create biased sampling effort, rather than a homogenous effort represented in our study, e.g., approximately all nights are sampled and included in our study (about 93 nights per radar per year). Third, we know that periods outside of the dominant bird migration window used in this study — typically, mid-June to mid-July — show much lower reflectivity measures than what we see at night during bird migration periods. Since this summer period coincides with peak activity for insects — as a worst-case situation — this comparison provides additional evidence that insect scattering is low compared to bird scattering during migration periods. Forth, Stepanian (2015) demonstrated the disproportionate relationship between scatterers of different sizes. Specifically, Stepanian (2015) wrote:

“Under the Rayleigh assumption, reflectivity is proportional to the product of the insect diameter to the sixth power and the number concentration. As a simple example, consider a 1-mm gnat and a 1-cm housefly. Based on the diameter component of reflectivity factor, the housefly contribution is six orders of magnitude greater than the gnat; or in other words, the reflectivity contribution from size is 1,000,000 times greater for the fly. Similarly, when a large size discrepancy exists between populations of scatterers, reflectivity-weighted measurements will be heavily biased toward larger organisms. In this scenario, gnats could only have a comparable reflectivity contribution to houseflies if their number concentration exceed that of houseflies by a factor of one million. This analogy can be extended further by considering a volume of 1-cm houseflies and 10-cm warblers, although the effects of resonance may lessen the disparity. In every case, the only way smaller organisms will have a significant contribution to the final signal is if their number concentrations are many orders of magnitude higher than the larger organisms.”

This relationship means that radar returns are biased toward larger scatterers, like birds. Because our approach uses approximately 93 scans averaged across the season, we are confident that the signals are dominated by migratory birds. However, we must acknowledge that an unknown proportion of our signal is surely composed of insects. The conditions that promote migration for birds are likely similar for the promotion of flight for nocturnal insects — likely the two taxa are migrating concurrently, but the radar measures in pulse volumes are highly skewed toward the large bird scatterers. We value the reviewer’s concern and don’t wish to offer a dismissive response to this important concern. To clarify and add transparency, we have added the following sentence to our methods section to provide clarity that contamination from insects is possible, but likely reflects a small bias:

*Lastly, in some regions of the southern United States, high density bat emergences from roosts also coincide with the onset of nocturnal bird migration. Because large bat roost departure events show unique ring-shaped features that are visually obvious (Stepanian and Wainwright 2018), we were able to screen for these signatures and generate buffers around roost locations and also permanently remove these areas from further analysis (set to NA). In all, we screened 12 sites (KAMA, KCRP, KDFX, KEOX, KEWX, KGRK, KHGX, KLBB, KMXX, KSJT, KTLH, KVAX); states included Alabama, Florida, Georgia, and Texas. One site, KEWX (San Antonio, Texas) was so severely contaminated by bats that we removed the station completely from our analysis, reducing the number of radars to 142. **Lastly, we note the insect classification remains a challenging task at the voxel scale within NEXRAD observations. However, because radar reflectivity is proportional to the product of the scatter diameter to the sixth power, measures strongly skew towards larger scatterers, like birds (56). Additionally, past studies on NEXRAD data, such as (57) showed no significant difference in models of bird migration intensity trained on data with or without insect dominated scans via airspeed filtering. For this reason, we are confident that our measures reflect signals from bird targets.***

2. Fig S1 shows how exodus time varies geographically, is there any pattern here? Is it affected by the level of light pollution?

This is a great question. To answer this question, we used data from Figure S1 to examine the relationship between exodus sampling times (minutes after local sunset) and mean skyglow per

radar sampling area at two scales (37.5 km and 80-km buffers). In preparing our response to this inquiry, we also noted that sampling time appeared to broadly follow variation in elevation. We assessed the influence of elevation and skyglow (predictor variables) on sampling time (response variable) using a simple linear regression model. Across the two spatial scales we assessed, we found that elevation was a significant predictor of sampling time ($p < 0.001$); however, skyglow was not a significant predictor of sampling time at either scale (37.5 km buffer, $p = 0.383$; 80 km buffer, $p = 0.751$). Across the contiguous United States, as elevation increased, the maximum rate of increase in radar reflectivity was later in the night. These models explained 51.7% (37.5 km scale) and 51.4% (80 km scale) of the variance in sampling time.

We have added the following text to support this statement:

Using these times, we explored drivers of variation in sampling time using two linear regression models with sampling time as the response variable and elevation and mean skyglow as predictors. We quantified mean skyglow at two scales, within 37.5 km and 80 km buffers; one model was run for each scale. At both scales, we found that radar site elevation was a significant predictor of sampling time ($p < 0.001$), but not mean skyglow (37.5 km buffer, $p = 0.383$; 80 km buffer, $p = 0.751$). These models explained 51.7% (37.5 km scale) and 51.4% (80 km scale) of the variance in sampling time.

3. Figure 1 – the dashed line is presumably the 1:1 relationship, the legend should say this; what about adding the regression lines, which probably will have a slope different from 1:1; why are the predicted values consistently lower than observed when density is high?

We have added a statement confirming that the dashed line reflects the 1:1 relationship between the predicted and the actual migrant density. We are happy to add the regression lines too if that is suggested.

Fig. 1: Scatterplot of predicted versus actual measured migration stopover density held out during model training from model sets. Points were held out from a 10-km bounding box within one of the randomly selected 400-km bounding boxes and one year was randomly selected, among the five possible years. The 10-km box was centered on the median X and Y coordinates of training points within the 400-km bounding box. Spring $R^2=0.85$ ($n=42,432$) and fall $R^2=0.87$ ($n=42,866$). **The dashed line shows a 1:1 relationship between predicted and actual migrant density.**

Regarding the consistent underprediction. This is a valuable observation and is in large part a product of using a tree-based machine learning algorithm. While our XGBoost models are parameterized to conduct a regression task, in many ways, the trees are generating a complex classification algorithm. Because the algorithm learns associations from the training data, it can only make predictions within the range of the response variable. This limitation is mechanistic, which often makes the algorithms poor at extrapolation beyond the sampling space seen in the training process. Additionally, large values are often underpredicted because they are not well represented in the training dataset (i.e., hotspots are rare across the sampling space) and, thus, the predictions bias towards the mean. It's possible that a multi-step model could assist in this challenging problem, however more methodological research is needed to investigate steps to alleviate underprediction. Similar patterns of underprediction were noted in Van Doren and Horton (2018, see Figure 3A). Additionally, Kuhn and Johnson (2013) note the following: “One limitation of simple regression trees is that each terminal node uses the average of the training set outcomes in that node for prediction. As a consequence, these models may not do a good job predicting samples whose true outcomes are extremely high or low.”

Kuhn, M., and K. Johnson (2013). Regression Trees and Rule-Based Models. In Applied Predictive Modeling (M. Kuhn and K. Johnson, Editors). Springer, New York, NY, pp. 173–220.

4. Line 111. The flyways (west, central, east) are not defined, they should be illustrated (see below)

Thank you for this comment. We did define the flyways in line 470 within the methods section as:

“Lastly, for summaries, we also examined migrant stopover density by flyway classification (73), in which the western flyway is defined as the contiguous United States west of 103° west longitude, the central flyway is the contiguous United States between 103° and 90° west longitude, and the eastern flyway is the contiguous United States east of 90° west longitude.”

We have added the flyway delineations to the following figures to add clarity and added our definition to the figure description. We believe that by making your suggested edits to Figure 4 now resolve any confusion.

*Fig. 4: Migratory bird stopover density and hotspot maps for the contiguous United States. (A) Spring and (B) fall predicted migrant stopover density for 2020. (C) Spring and (D) fall relative stopover categories from predicted 2020 stopover density. Red shades denote pixels above the 90th quantile of predicted stopover density, yellow pixels between the 50th and 90th quantile of migrant stopover density, and gray showing pixels below the 50th quantile of migrant stopover density. Relative quantiles identified using a circular focal window radius of 265 km, which relates to measured average nightly flight distances of tracked free-flying Swainson’s (*Catharus ustulatus*) and hermit (*C. guttatus*) thrushes (Wikelski et al. 2003). **The western flyway is defined as the contiguous United States west of 103° west longitude, the central flyway is the contiguous United***

States between 103° and 90° west longitude, and the eastern flyway is the contiguous United States east of 90° west longitude.

Fig. 5: Seasonal difference in predicted migrant stopover density (Fall minus spring stopover density). Blue shades show greater stopover activity in the fall, red greater activity in the spring, and white showing no seasonal difference.

5. Lines 129-138. The text here needs to be related to a figure, which I suspect is Figs 4A & B. Please cite. Also Fig 4 should illustrate the flyway limits, and I think that box plots could be added to show the density values per flyway that is discussed in this section.

We appreciate your suggestions to make the flyways clearer and to better visualize variations in densities across the flyways. We have changed Fig 4A and B to match your suggestions.

6. As Fig 3 not mentioned yet, the order of the figures should be changed. In fact, Fig 3 is not really discussed anywhere – what is it showing? Needs to be documented in the results.

Thank you for bringing this to our attention. We have reordered Figures 3 and 4 to better match the order of our text. The text now reads as follows:

During spring, we found the central portion of the country showed the greatest stopover densities — in fact, on average, migrant stopover density within this region was 1.5 times greater in the central flyway as compared to the eastern flyway, and 2.9 times greater as compared to the western flyway (Fig. 3A). In the spring, Arkansas, Oklahoma, Louisiana, Texas, and Mississippi, showed the greatest mean stopover density, in descending order. The highest stopover densities were in the coastal Gulf of Mexico region, particularly southern Texas. In the fall, stopover density

was greatest in the southeastern United States, with Alabama, Tennessee, Arkansas, Mississippi, and Georgia showing the greatest mean stopover density, in descending order. The greatest fall stopover density resided in the eastern flyway, showing 1.2 times more than the central and 5.8 times more than the western flyway (Fig. 3B).

We also generated relative focal stopover maps parameterized by average songbird flight distances (265 km (Wikelski et al. 2003)), with three levels of varying intensity (Fig. 3C and 3D). Broadly, hotspots often resided near coastlines, geographic barriers (e.g., mountain ranges in Colorado and California), and in regions with large swaths of forest. However, these represent generalizations, and each region showed a specific combination of drivers of stopover density (Fig. 4).

7. Line 152 – I don't think "tantamount" (meaning equivalent to) is the right word to use here.

Thank you for your comment. We have changed tantamount to paramount so that now the sentence reads as follows:

*"Stopover locations are **paramount** to the passage of billions of migratory birds. For decades, identifying and prioritizing stopover locations has remained a scientific priority— we fill a perennial gap by providing the quantification of high density stopover locations."*

Jason Chapman

Reviewer #2 (Remarks to the Author):

The authors of the manuscript titled "Reshaping avian migration in the Anthropocene — continental scale attraction to artificial light at night revealed" provided a comprehensive analysis of bird stopovers during spring and fall migration based on five years of data, over the contiguous United States. The analysis generated maps of migratory bird density (spring and summer) and identified positive and negative relationships of bird density to landscape characteristics. The skyglow variable, quantifying light pollution, had the second strongest positive effect on bird density, after elevation. This result is concerning for bird conservation, as collision with urban structures is a known cause of bird mortality during migration. The manuscript is clear and concise and the figures are representative of the main findings of the manuscript.

The data processing and analyses are robust and well-documented, thus the results and conclusions presented in this manuscript are reliable. While I don't have experience with radar data, the analytical steps the authors provided enough details to understand the radar data processing. The modeling component (gradient boosted trees) was also well-documented and designed. I don't have any suggestions for improving the models. I think this study provides novel and valuable insights into spatial patterns of migratory bird density at stopovers that can inform bird conservation and land management.

I only have a few minor editing suggestions.

Line 38: revise "may act as ecological traps at macroscales, bringing migrants into dangerous, modified habitats." as "may act as ecological traps at macroscales that increase mortality of birds"

during

migration”

Thank you for this suggestion. We have now changed the abstract to read as the following:

“This finding points to an expanding threat to avian migrants: peri-urban illuminated areas may act as ecological traps at macroscales that increase mortality of birds during migration”

Line 96: reword as “amassed 2,000 models at 400 km resolution across the contiguous US”

We have changed the sentence to match your suggestion. It now reads as:

“Using the US NEXRAD network, we harnessed the hierarchical spatial structure of stopover measures from more than 1 million locations, assembled 49 predictors, and amassed 2,000 models at 400 km resolution across the contiguous United States to provide the first view of continent-wide migration stopover.”

Line 181: references 34 and 35 don’t seem to be studies about the relations between vegetation structure and bird diversity and abundance. A Google Scholar search “bird diversity vegetation structure” returned papers that may provide a stronger support for this statement, for example:

Thank you for the suggested references. Our existing citations described functionality of GEDI and the datasets produced. We have included the following citations to provide specific examples of the relationship between vegetation structure and bird diversity and abundance:

Bakx, T.R., Koma, Z., Seijmonsbergen, A.C. and Kissling, W.D., 2019. Use and categorization of light detection and ranging vegetation metrics in avian diversity and species distribution research. Diversity and Distributions, 25(7), pp.1045-1059.

Carrasco, L., Giam, X., Papeş, M. and Sheldon, K.S., 2019. Metrics of lidar-derived 3D vegetation structure reveal contrasting effects of horizontal and vertical forest heterogeneity on bird species richness. Remote Sensing, 11(7), p.743.

Farwell, L.S., Gudex-Cross, D., Anise, I.E., Bosch, M.J., Olah, A.M., Radeloff, V.C., Razenkova, E., Rogova, N., Silveira, E.M., Smith, M.M. and Pidgeon, A.M., 2021. Satellite image texture captures vegetation heterogeneity and explains patterns of bird richness. Remote sensing of Environment, 253, p.112175.

Line 406: provide the spatial resolution of this product.

We have included this suggestion. The sentence now reads as follows:

“We used the Visible and Infrared Imaging Suite (VIIRS) Day Night Band (DNB) monthly cloud-free DNB composite products. These data products, produced by the Earth Observation Group, Payne Institute for Public Policy, are at a 15 arc second, or roughly 500m resolution. These products remove non-stable lighting from the imagery.”

Lines 757-758: revise here to remove the double negative. Possible revised text: “Note that neutral indicates that the variable was not dominantly positive or negative across all models.”

Thank you for your comment. We have changed the text to match your suggestion:

“Note that neutral indicates that the variable was not dominantly positive or negative across all models. However, it could hold a negative or positive value.”

Reviewer #3 (Remarks to the Author):

The authors present a novel analysis of factors affecting avian migrations in the United States using data collected from a network of 142 Radars. They include a large number of potential explanatory terms in a niche model including the enhanced vegetation index (EVI), land cover classes and composition, percent canopy cover, percent impervious surface, accumulated nocturnal degree-days, precipitation, skyglow, elevation, distance to radar, and year. The authors identified skyglow as a highly influential and consistently positive predictor of bird migration stopover density in 70% of their models.

The paper is very interesting, well written and brings together multiple large remote sensing data sources from across different platforms. I feel the paper would make a nice contribution to Nature Communications and should be highly cited.

I do however have some questions I would like to see clarified before the manuscript can be considered for publication.

The response data (bird stopover density) is derived from a network of 142 radar stations, each sampling a very small region (250-m x 0.5° x 1.0°). My immediate concern is then that the model outcomes and the authors interpretations of them are highly dependent on the locations of these 142 radar stations in relation to the spatial distribution of the predictors. 49 spatial predictors for a response that is replicated in 142 unique spatial locations is very small replication indeed.

Thank you for allowing us to expand on the spatial distribution of the predictor variables. We wanted to clarify that our bird stopover density measures, in our eyes, are based on a large percentage of the sampling area highlighted in this study. While it is true that we use 142 radars in our study, each radar collects many unique values within the sampling region (upwards of 13,000 unique pixels within an 80km sampling region). In total, across the US, the network of 142 radars sampled 1,002,511-km² locations used in our models.

Specifically, we use 1,002,511 random points in our model training (of 1,985,170 possible locations). In total, the radars sampled 26.2% of the contiguous United States. From these random points, we sampled migrant stopover density across five years during both spring and fall. We hope that clarifies this concern.

In total, we included 49 predictor variables in our modeling of migrant stopover density. We extracted 1,002,511 random points from the radar coverage area (50.5% of possible locations) and ensured that no single location was replicated in our training dataset (e.g., checking that

multiple random points did not fall within a 1 km pixel). Our radar coverage area represented 26.2% of the contiguous United States; distributional modeling filled the remaining 73.8% of the land area. We used gradient boosted trees, carried out through the XGBoost (Chen and Guestrin 2016, Chen et al. 2017) package in R, to examine relationships between predictor variables and stopover density. This approach uses a tree ensemble model, which consists of a set of regression trees, as applied in a supervised learning environment that relates a training dataset to a response variable (i.e., stopover density). We divided our dataset into three groups: a training set (75%) for learning; a validation set for model tuning (15%); and a test set to evaluate performance (10%). We randomly assigned locations to these categories to ensure independence across the sets at the location level.

I have not seen any description of the predictor variable conditions at each radar station, or any evidenced discussion of how reflective this is of the distribution of these predictors across the wider US. If radar station are predominantly suburban for example, then their may be considerable bias in the dataset, particularly for those located in arid environments. The end result might be that birds tend to stop over where there is more water or forage, and this happens to be locations where radar stations tend to be located, and also where there are military installations and hence more skyglow.

I'm not saying that this is the case, rather that the authors have not evidenced that it is not the case. A simple presentation of the response variable conditions at each radar station compared to the range of those responses across the US should help (hopefully) rule out any source of bias in the analysis.

This is a good point made by the reviewer. We hope that our response to the previous comment helps add some clarification regarding how the radars are sampling and positioned. However, the macroscale positioning of the radars is likely more biased towards urban areas — the radars are strategically positioned to sample meteorological events near human population centers. For this reason, we sought to provide a statistical description of the covariation of some of the critical stopover variables.

We examined the correlation of our predictor variables, including correlations between skyglow and % canopy cover, skyglow and % summed forest cover types (NLCD classes 41, 42, and 43), skyglow and % open water (NLCD class 11), and just as a proof-of-concept, % canopy cover and % summed forest cover types, which we predicted would be highly correlated. For habitat specific correlations, we focused on the 5 km buffer scale, since these predictors ranked highly in our models (as opposed to the 1 km scale). Additionally, for these correlations, we used one season, 2016, because % canopy cover only has one annual replicate — otherwise, in our correlations, pixel values would be pseudo-replicated (i.e., the same value repeated in 2017–2020). Lastly, rather than examining correlations across all possible pixels (~1 million), we examined the Pearson's correlation coefficient on 100 randomly drawn locations and did this 10,000 times. We did this because of the relationship between sample size and statistical significance. For each correlation, we captured the correlation coefficient and the p-value. We then summarized the distribution of correlation coefficients and proportion of significant correlations, based on an alpha value of 0.05. We show these results in Figure S4. Generally, we only found very weak correlations between skyglow and canopy cover, forest cover types, and open water (median values between -

0.1–0.04), with between 1.2% and 10.2% of correlations showing significance. Unsurprisingly, we found strong positive correlations between %canopy cover and % summed forest cover types, with 100% of correlations showing significance.

We have added the following figure to our supplemental materials.

Figure S4: Pearson's correlation strength between model predictors, including (A) skyglow and % canopy cover, (B) skyglow and proportion of NLCD forest cover types (41, 42, and 43) within a 5 km buffer, (C) skyglow and proportion of NLCD open water within 5 km buffer, and (D) % canopy cover and proportion of NLCD forest cover types (41, 42, and 43) within a 5 km buffer. Correlations were conducted on 10,000 random selections of 100 locations from spring of 2016. Significance based on an alpha value of 0.05.

And we added the following text to our methods section:

Correlation of predictors

When examining the influence of model predictors, we wanted to understand the relationship between a suite of core predictors, including skyglow and habitat cover types. While we predict skyglow, forest cover, and riparian corridors to have a positive association with

stopover density, we wanted to understand if their occurrence was correlated (e.g., is high skyglow also associated with a high percent of canopy cover?). To test this, we summarized the distribution of correlation coefficients and proportion of significant correlations (based on an alpha value of 0.05), examining pairwise correlations of skyglow and % canopy cover, skyglow and proportion of forest cover types (NLCD classes 41, 42, and 43), skyglow and proportion of open water (NLCD class 11), and just as a proof-of-concept, % canopy cover and proportion of forest cover types (NLCD classes 41, 42, and 43), which we predicted would be highly correlated. For habitat specific correlations, we focused on the 5 km buffer scale. We used predictors from one season, 2016, because % canopy cover only had one replicate in our study — otherwise, in our correlations, pixel values would have been pseudo-replicated (i.e., the same value repeated in 2017–2020). Lastly, rather than examining correlations across all 2016 pixels (~1 million), we examined the Pearson’s correlation coefficient from 10,000 random draws of 100 locations. We show these results in Fig. S4. Generally, we found very weak correlations between skyglow and canopy cover, skyglow and forest cover types, and skyglow and open water (median Pearson’s correlation coefficient values between -0.1–0.04), with between 1.2% and 10.2% of correlations showing significance. We found strong positive correlations between % canopy cover and proportion of forest cover types, with 100% of correlations showing significance.

Reviewers' Comments:

Reviewer #1:

Remarks to the Author:

The authors have done a thorough and convincing job of replying to the comments I made on the first round of revision, and editing the revised manuscript accordingly; after reading the paper through again, I have no further comments. I believe this paper makes a fine contribution to the field and is now ready for publication.

Jason Chapman

Reviewer #3:

Remarks to the Author:

The authors have presented an impressive and detailed rebuttal to the reviewers comments. With regards to my specific questions around low sampling area/replication and potential for collinearity between predictors they have provided ample evidence that neither should be considered a major concern for this analysis. While one or two predictors are collinear, the skyglow predictor is not.

I note that the authors have opted to reframe the title of the manuscript response to the comments of reviewer 1. Personally I feel this is a little over-conservative but will leave that to the discretion of the editor.